# Pediatric Onset Multiple Sclerosis and Obesity: Defining the Silhouette of Disease Features in Overweight Patients

**DOI:** 10.3390/nu15234880

**Published:** 2023-11-22

**Authors:** Laura Papetti, Elena Panella, Gabriele Monte, Michela Ada Noris Ferilli, Samuela Tarantino, Martina Proietti Checchi, Massimiliano Valeriani

**Affiliations:** 1Developmental Neurology Unit, Bambino Gesù Children’s Hospital, IRCCS, 00165 Rome, Italy; gabriele.monte@opbg.net (G.M.); michela.ferilli@opbg.net (M.A.N.F.); samuela.tarantino@opbg.net (S.T.); martina.proietti@opbg.net (M.P.C.); massimiliano.valeriani@opbg.net (M.V.); 2Child Neurology and Psychiatry Unit, Systems Medicine Department, Hospital of Rome, Tor Vergata University, 00133 Rome, Italy; elena.panella@opbg.net; 3Center for Sensory Motor Interaction, Aalborg University, DK-9220 Aalborg, Denmark

**Keywords:** pediatric onset multiple sclerosis, obesity, onset, risk factor

## Abstract

Obesity has been suggested as an environmental risk factor for multiple sclerosis (MS) and may negatively effect the progression of the disease. The aim of this study is to determine any correlation between overweight/obesity and the clinical and neuroradiological features at the onset of pediatric onset multiple sclerosis (POMS). Were included patients referred to the POMS Unit of the Bambino Gesù Children’s Hospital between June 2012 and June 2021. The diagnosis of MS with an onset of less than 18 years was required. For all included subjects, we considered for the analysis the following data at the onset of symptoms: general data (age, sex, functional system compromised by neurological signs, weight and height), brain and spinal magnetic resonance imaging (MRI), cerebrospinal fluid exams. We identified 55 pediatric cases of POMS and divided them into two groups according to the body mass index (BMI): 60% were healthy weight (HW) and 40% were overweight/obese (OW/O). OW/O patients experienced a two-year age difference in disease onset compared to the HW patients (12.7 ± 3.8 years vs. 14.6 ± 4.1 years; *p* < 0.05). Onset of polyfocal symptoms was seen more frequently in OW/O patients than in HW (72.7% vs. 21.2%; *p* < 0.05). The pyramidal functions were involved more frequently in the OW/O group than in the HW group (50% vs. 25%; *p* < 0.005). Black holes were detected more frequently in OW/O patients in onset MRI scans compared to the HW group (50% vs. 15.5%; *p* < 0.05). Our findings suggest that being overweight/obese affects the risk of developing MS at an earlier age and is associated with an unfavorable clinical–radiological features at onset. Weight control can be considered as a preventive/therapeutic treatment.

## 1. Introduction

Multiple sclerosis (MS) is a chronic inflammatory autoimmune disease of the central nervous system (CNS) that is most commonly diagnosed in young adults; 3–5% of patients have disease onset under the age of 18, and less than 2% of patients are younger than 10 years [1,2,3,4]. Pediatric MS has some differences from adults in the clinical presentation: the pediatric population presents a higher relapse rate, despite better recoveries after relapses, and a slower evolution toward a secondarily progressive disease [1]. About half to two-thirds of pediatric patients present with multiple symptoms [4].

Obesity is a major public health problem, as the prevalence of obesity in Italian school-aged children is 9.4%, while the prevalence of overweight is 20.4% [5]. Furthermore, the prevalence of overweight and obesity persists, with an increasing trend, over the years, following the trend of MS diagnosis in children [6].

The etiology of MS is still unknown, although current evidence suggests that it is the result of autoimmune injury triggered by a combination of an environmental stimulus in genetically susceptible individuals, both adults and children. Low serum vitamin D levels, passive or active smoking, remote infection with Epstein–Barr virus, and obesity are factors that may increase environmental risks [6,7,8].

Adipose tissue, which is involved in many metabolic functions, regulates the immune system and endocrine function. These functions are mediated by various components of adipose tissue: adipocytes, which exert an endocrine function through the secretion of various adipokines (adiponectin, leptin, and resistin), innate and adaptive immune cells (macrophages, neutrophils, eosinophils, mast cells, and various T and B cells), and fibroblasts. This is closely connected to how adipose tissue regulates inflammatory status [5]. An excessive amount of adipose tissue leads to an increased secretion of inflammatory substances, such as pro-inflammatory cytokines like tumor necrosis factor-α (TNF-α), monocyte chemoattractant protein-1 (MCP-1), and interleukin-6 (IL-6); at the same time, the secretion of anti-inflammatory adipokines is decreased in the over-weight/obese body. Obesity is associated with chronic inflammatory status as a result of these factors [5]. Indeed, obesity in adults is associated with increased cerebrospinal fluid (CSF) levels of proinflammatory molecules, including IL -6 and leptin, and a de-crease in anti-inflammatory mediators such as IL-13 [9,10]. Similar data have been found in the pediatric population, particularly in prepubertal patients, suggesting an interaction between excess body fat, sex hormones, and the occurrence of pediatric onset multiple sclerosis (POMS) [11]. An hypothesis is that children are more susceptible to inflammatory damage due to blood–brain barrier (BBB) permeability, resulting in more pronounced acute axonal damage in inflammatory demyelinating lesions than in adults [12], with a greater number and volume of new T2 lesions on brain magnetic resonance imaging (MRI) [13], which may affect outcomes, especially if they are abundant on baseline neuroimaging [14,15]. It is particularly relevant to identify the risk factors involved in the development of the disease during the early stages of life, considering this time as a window to potentially interfere: in this sense many efforts have been spent and some studies focused on the role for childhood obesity in risk of developing MS.

The presence of high BMI values in pediatric multiple sclerosis patients in their childhood and before the onset of neurological symptoms is a common occurrence [6,11,16,17]. Despite considering environmental and genetic risk factors, the connection between high BMI and increased risk of MS remains valid [18]. Regarding genetic factors, a recent study performing mendelian randomization could confirm the positive causal association between high BMI and MS, and detected as well common risk genes shared between MS and obesity, suggesting a potential pathogenetic mechanism to justify this comorbidity [16]. High BMI may be considered not only a risk factor but also a negative predictive fac-tor, as it appears to be associated with higher rates of relapse [10], development of disability [19], and negative response to disease-modifying drugs in terms of relapses under treatment [20]. Meanwhile, as evidence increases about this correlation, the prevalence of obesity in the MS pediatric population is growing at a rate of 25–50% compared to the past [21].

The objective of this research is to examine whether overweight/obesity is associated with the clinical and neuroradiological presentation of POMS.

## 2. Materials and Methods

### 2.1. Participants and Inclusion Criteria

We performed a retrospective study that includes patients that referred to the POMS Unit of the Bambino Gesù Children’s Hospital, between June 2012 and June 2021. The 2013 International Pediatric MS Study Group (IP-MSSG) criteria were used to diagnose MS, and these patients were included in the analysis [22]. Pediatric MS diagnosis was made based on these criteria if the patient had a history of:(1)Two or more non-encephalopathic clinical CNS events with presumed inflammation, separated by more than 30 days, involving more than one area of the CNS;(2)One non encephalopathic episode typical of MS which was associated with MRI findings consistent with 2010 Revised McDonald criteria for dissemination in space (DIS) and in which a follow up MRI shows at least one new enhancing or non enhancing lesion consistent with dissemination in time (DIT);(3)One ADEM attack followed by a non encephalopathic clinical event, three or more months after symptom onset, that was associated with new MRI lesions that fulfill 2010 Revised McDonald for DIS criteria;(4)A first, single, acute event that does not meet ADEM criteria and whose MRI findings are consistent with the 2010 Revised McDonald criteria for DIS and DIT (this last criterion was valid only for children over 12 years of age.

Other inclusion criteria for the studied are: age younger than 18 years at clinical onset; the other inclusion criteria included: availability of an examination of the liquor taken by rachicentes (physical chemical examination, determination of oligoclonal bands on liquor and serum and microbiological examinations); examination of cerebral and spinal MRI with and without injection of gadolinium and carried out 30 days after possible steroid therapy and without the initiation of therapy modifying the course of disease.

Patients with other intercurrent chronic pathologies (endocrine, tumoral, cardiac, respiratory) that could affect BMI were excluded.

The study was approved by the Ethical Committee of Bambino Gesù Children’s Hospital (date 19 October 2023). All patients enrolled and their parents provided consent for the publication of the results.

### 2.2. Data Collection

Medical records performed at the moment of disease onset, before starting steroid treatment and disease modifying therapy were retrospectively reviewed for each patient. The collected data included: (1) demographic variables: sex, age; (2) clinical variables: measurement of weight (kg), height (cm), BMI calculated as weight in kilograms divided by height in meters squared, expanded disability status scale (EDSS) score, presenting clinical symptoms; (3) laboratory data: presence of oligoclonal bands (OCBs) in CSF, determined by isoelectric focusing, combined with immunoblotting of matched serum, and CSF sample pairs, presence of pleocytosis in CSF, defined as >5 white blood cells/mm^3^, previous Epstein–Barr virus (EBV) infection, defined by measuring serum viral antibodies (IgM and IgG by ELISA) and performing quantitative real-time PCR for the EBV; (4) characteristics of brain and spine MRI, performed at the time of clinical onset with a 3T scanner, acquiring axial and sagittal T2-weighted, fluid-attenuated in-version recovery (FLAIR)- weighted, T1-weighted MRI sequences, and T1-weighted MRI images after administration of gadolinium.

Clinical symptoms were later classified according to the involvement of various functional systems, related to nervous system activity. In details the following symptoms or signs had been considered: Visual function (visual acuity, visual fields, and scotoma); Brainstem Functions (extraocular movement impairment, nystagmus, trigeminal damage, facial weakness, hearing loss, dysarthria, dysphagia, and other cranial nerve functions); Pyramidal functions (reflexes, limb strength, and spasticity); Cerebellar Functions (head tremor, truncal ataxia, tremor or dysmetria of limbs, and gai ataxia); Sensory Functions (superficial sensation of trunk and limbs, vibration sense of limbs, position sense of limbs, and paranesthesia); Bowel and Bladder Function (urinary hesitancy or urgency, and bowel disfunction).

We ensured that the brain MRI, performed at the onset, included axial and sagittal T2-weighted, fluid-attenuated inversion recovery (FLAIR)-weighted, T1-weighted MRI sequences, and T1-weighted MRI images after administration of gadolinium. All patients included in the study also underwent a spinal MRI at the onset of symptoms and before the start of corticosteroid therapy; dual-echo (proton-density and T2-weighted) conventional and/or fast spin-echo, STIR (as alternative to proton-density weighted) and contrast-enhanced T1-weighted spin-echo (in case of presence of T2 lesions) sequences were acquired. The MRI scan revision was centralized and carried out by two operators (LP and MANF) blinded to clinical outcome, at the Bambino Gesù children’s hospital. Lesion characteristics were recorded, including the location, distribution, border outline, symmetry, and number, as well as size and gadolinium capture. Tumefactive lesions were defined as such if larger than two cortical gyri. The presence or absence of black holes (non-enhancing hypointense lesions visible on T1-weighted sequences) and post-gadolinium enhancement were analyzed.

### 2.3. Subgroups Classification

The population was divided in two groups, according to the BMI, using the Center for Disease Control (CDC) metrics [6]: the first group (healthy weight, HW) includes people with BMI under 25, and the second group (overweight/obese, OW/O) included people with BMI of 25–29.9, considered overweight, and people with a BMI over 30, categorized as obese.

### 2.4. Statistical Analysis

We used the chi-squared test to compare the distribution of categorical variables in the two groups. The categorical variables included sex, presence of encephalopathy at clinical onset, symptoms at onset in various functional systems, presence of oligoclonal bands in CSF, distribution of lesions in MRI in the different areas (periventricular, subcortical, subtentorial, and spine), presence of gadolinium-enhancing lesions, black holes and swelling lesions.

The Mann–Whitney U-test was used to compare continuous variables like age, the number of the relapses before diagnosis and the number of lesions visible in MRI scans taken at onset.

Statistical analysis was conducted with SPSS software version 22. The level of significance was set for *p* value values lower than 0.05.

## 3. Results

Among 65 patients selected, 10 of them were excluded for unavailability of data required (3 no onset MRI images available, 2 no data from CSF collected at onset available, 5 no measurement of weight and height performed at onset).

The statistical analysis included 55 patients with a diagnosis of relapsing-remitting MS with pediatric onset. This population was composed of 34 girls (61.8%) and 21 boys (38.2%), with a mean age of 13.5 years (Table 1).

Patients were divided into two groups: HW (33 patients, 60%) and OW/O (22 patients, 40%). The HW group consisted of 57.6% girls and 42.4% boys while the OW/O population was composed of 68.2% girls and 31.8% boys (*p* > 0.05) (Table 2).

### 3.1. Age at Disease Onset

OW/O patients had an age of disease onset approximately two years lower than HW patients (12.7 ± 3.8 years vs. 14.6 ± 4.1 years; *p* < 0.05).

### 3.2. Clinical Features at Disease Onset

An acute disseminated encephalomyelitis (ADEM)-like with encephalopathy at onset occurred with a similar frequency in the OW/O group versus HW patients (*p* > 0.05). Excluding patients with ADEM-like, an onset with polyfocal symptoms was more often seen in OW/O patients than in the HW group (72.7% vs. 21.2%; *p* < 0.05). In order of frequency, the onset symptoms occurred in patients with polyfocal presentation involved superficial/proprioceptive sensation (No. 16, 69.5%), brainstem deficit (No. 12, 52.1%), pyramidal functions (No. 12, 52.1%), cerebellar functions (No. 11, 47.8%), visual deficit (No. 8, 34.7%), bowel and bladder functions (No. 2, 8.6%).

On the contrary, monofocal onset was seen more often in HW patients than in the OW/O group (66.6% vs. 18.1%; *p* < 0.05). In order of frequency, the onset symptom occurred in patients with monofocal presentation involved visual deficit (No. 9, 34.6%), brainstem deficit (No. 7, 26.9%), superficial/proprioceptive sensation (No. 6, 23%), pyramidal function (No.4, 15.3%).

Involvement of the pyramidal functions was more often detected in the OW/O group than in the HW group (50% vs. 25.4%; *p* < 0.005), (Table 3). The analysis of the total number of relapses that occurred before diagnosis did not reveal a statistically significant difference between the two groups.

### 3.3. MRI Features at Disease Onset

Regarding MRI scans performed at onset, a statistically significant difference was revealed analyzing the number of black holes: they were more frequently detected in OW/O patients in onset MRI compared to the HW group (50% vs. 15.5%; *p* < 0.05) (Figure 1).

When analyzing the presence of other MRI features such as periventricular lesions, juxtacortical/cortical lesions, infratentorial lesions, spinal cord lesions, gadolinium-enhanced lesions, and tumefactive lesions, no statistically significant differences were found between the two groups (Table 4).

We found no statistically significant difference between the two groups about the total number of T2 hyperintense cerebral lesions at onset.

### 3.4. Laboratorial Features at Disease Onset

Oligoclonal bands (OCBs) in CSF were found in 26 HW patients (83.9%) and in 19 (95%) OW/O patients, pleocytosis were found in 13 HW patients (50%) and in 14 OW/O patients (70%), previous EBV infection were found in 30 HW patients (91%) and in 22 OW/O patients (100%).

We found no statistically significant difference between the two groups analyzing the presence of oligoclonal bands (OCBs) in CSF, the presence of pleocytosis in CSF, previous EBV infection.

## 4. Discussion

Our study shows that patients with POMS who are OW/O experience clinical onset at a younger age than those who are HW. At the onset of the disease, OW/O individuals exhibit a worse clinical picture and less favorable MRI findings than HW individuals. POMS is a disease that is caused by genetic and environmental factors, which increase the risk of development [23].

Susceptibility to SM has been identified in several genes within the major histocompatibility complex (MHC) loci. Around one-third of them have been associated with POMS, which suggests that there is a shared genetic inheritance. The HLA-DRB1 gene is the gene responsible for the most significant genetic contribution, which is associated with changes in HLA in general and specifically [24].

Epstein–Barr virus (EBV) infection is a determinant of POMS that has been extensively studied [25,26]. Vitamin D deficiency, not breastfeeding infants, pesticide exposure, smoking, air quality, and hormonal influences [18] were among the other factors [18]. Furthermore, certain risk factors seem to have a more significant impact during a particular time frame. BMI and obesity in adolescence, not during childhood, is associated with an increased risk of developing MS [27,28,29,30].

Authors, in particular, attempted to establish if there was a correlation between obesity age and the risk of MS [17,23]. Identifying the age range in which obesity may increase the future risk of developing MS is important for implementing prevention measures for being overweight. Some studies showed that in obese subjects, correcting their body weight reduces the risk of developing MS during the life period [17,27].

Recently a Mendelian randomization study was performed to evaluate whether childhood BMI has a causal influence on MS, and whether this putative effect is independent of early adult obesity and pubertal timing. This study found that a higher genetically predicted childhood BMI was associated with increased odds of MS. The association between childhood obesity and MS susceptibility was mediated by the persistence of obesity into early adulthood, but independent of the timing of puberty [31].

A Danish longitudinal study conducted in school children found that among girls, at each age from 7 to 13, a one-unit increase in a the BMI z-score was associated with significant hazard ratios of MS. The risk of MS increased by 1.61–1.95 times for girls in the 95th percentile for BMI compared to girls in the 85th percentile. The associations were weaker in boys. A hazard ratio of 1.17 was found for a one-unit increase in BMI z-score at age 7, and 1.15 was found at age 13 [17].

Munger et al. reported a study in which some women were questioned, using a self-reported representative pictogram about their body silhouettes at the ages of 5, 10, and 20. In this study, the author found that the women who reported having a larger body size at age 20 had a two-fold increased risk of MS compared to women who reported a thinner body size. There was also a suggestion that having a larger body size during childhood at ages 5 or 10 may increase the risk of MS. Furthermore, after adjusting for body size at age 20, there was no increased risk of MS associated with having a large body size during childhood. The twofold risk of MS associated with large body size at age 20 remained unchanged [28].

These studies support the previous findings that overweight individuals in late adolescence/early adulthood have a 40% increased risk of MS [32]. Unfortunately, we do not have information about the BMI trajectory of our cohort, but our data show that a high BMI at the onset of MS is associated with an earlier age of onset of MS, polyfocal symptoms at onset, and early hypointense MRI lesions.

The onset of MS in childhood has significant implications for the prognosis. Compared to adults, children and adolescents with MS experience a higher relapse rate and more commonly affect the cerebellar and brainstem regions [33]. Pediatric MS patients take about 10 years longer than adults to reach irreversible levels of disability. Nevertheless, these levels are attained by a final age that is 10 years younger than in adult-onset patients [34]. Furthermore, POMS can impact the cognitive function and development of children. Early-onset MS patients have a faster decline in cognitive performances compared to patients with adult-onset disease, resulting in a higher risk of cognitive impairment and psychiatric comorbidity in adulthood, even when adjusted for disease duration [35,36,37,38,39,40]. POMS is linked to psychiatric comorbidity in adulthood. Overall, it is estimated that approximately 25–30% of patients with POMS experience mild cognitive changes [40], mainly related to attention and processing speed [41]. Long-term repercussions on the cognitive performance of POMS patients have also been described [36]. Therefore, efforts towards early diagnosis, the discovery of early predictors of long-term outcomes, and appropriate early drug intervention are highly warranted.

Regarding clinical features at onset, a recent review estimated that approximately half to two-thirds of pediatric MS patients have a polysymptomatic presentation [42] and this is more frequent in patients with onset at a younger age, which may reflect a greater susceptibility of an immature brain to the inflammatory insult [43]. Children are most commonly diagnosed with motor dysfunction (30%), sensory symptoms (15–30%), brainstem symptoms (25%), optic neuritis (10–22%), and ataxia (5–15%) [38]. The results of our analysis show that a clinical onset with polyfocal signs is more frequently observed in overweight/obese patients than in healthy weight patients, who tend to present with monofocal clinical manifestations. In addition, in our study, overweight/obese patients are more likely to have pyramidal domain involvement (50% OW/O) than healthy weight patients (25.4% HW).

The presence of polyfocal symptoms at the clinical onset of MS has been associated in both children and adults, with an increased risk of moderate or severe disability [2,44,45,46] and a decreased response to disease-modifying treatments [47]. A higher EDSS score is linked to the involvement of the pyramidal system [48]. Langer-Gould et al. analyzing a multiethnic population of 75 new diagnoses of pediatric clinically isolated Syndrome (CIS) and POMS, found that moderately and extremely obese children were more likely to present with motor/sensory symptoms of transverse myelitis [16].

Regarding MRI findings at the onset of the disease, our analysis shows that hy-pointense T1 lesions (also known as Black Holes) [49], are found more frequently in the overweight/obese group than in the healthy weight group. The presence of black holes is a sign of chronic inflammatory damage and tissue damage due to axonal loss [50].

It is known in adults that the black hole burden is related to cerebral atrophy [51,52,53,54], and both reflect and are a negative prognostic factor of physical disability [49,55,56] and poor cognitive performance [57]. Furthermore, children with MS are more likely to present black holes already at onset of the disease compared to AOMS, probably due to a more aggressive disease early on and the susceptibility to axonal damage [12], and tend to have more rapid loss of brain parenchyma during the course of the disease [58].

Studies in the adult MS population on the effects of over weight on neuroradiological factors that are associated with progression and persistent disability have yielded conflicting results [10,59,60,61,62]. Some studies have documented a relationship between an increase in BMI and a reduction in cerebral gray matter over time, which leads to a greater burden of T1 lesions [59,61,62]. This type of relationship [10,60] has not been found by others, however. Other factors, such as the duration of the disease, may affect the reduction in brain volume or lesion load in the adult population [53].

The study of these phenomena in the pediatric population allows us to analyze the disease at an earlier stage, which therefore gives more value to the effect of weight. Few studies have investigated this topic in POMS, and those that have, have found that patient BMI did not affect the probability of presenting numerous T2 lesions or contrast enhancing lesions at the onset [63]. However, a second study did not observe significant differences in the clinical characteristics of POMS between normal weight and overweight subjects [20].

The correlation between overweight and MS can also have implications in the therapeutic field. In adults with MS, the risk of developing persistent disability has gradually decreased through the early use of effective disease-modifying treatments [64,65]. However, the therapeutic management of POMS has greater limitations than that of adults because we have fewer pharmacological strategies available than in adults [66,67]. Therapeutic targets can be focused on preventing and controlling overweight. Prior studies have demonstrated the protective effect of vigorous physical exercise during adolescence and childhood against the development of MS [14,68,69] and the protective effect of physical exercise on disease outcome, in terms of accumulation of disability, relapse rate, neurocognitive performance and MS-related MRI lesions accrual [70].

Diet may also play a role in susceptibility to developing MS and may affect the course of the disease in both adults [71,72] and in pediatric populations with MS [73,74]. In recent years, MS treatment strategies have included other therapeutic dietary interventions, in addition to vitamin D supplementation [53]. Omega-3 and omega-6 acid supplements [75] and the ketogenic diet [76,77] are included. Dietary interventions should be considered as a potential therapeutic strategies that may impact pathophysiological mechanisms and the well-being of patients with MS.

In conclusion, although our study highlighted and presented more evidence about how negatively obesity influences the onset features of POMS [53], it is possible, even after the disease onset, to remove risk factors, ameliorate the progression of disability and comorbidities, and improve the neurological reserve, able to repair and compensate for neuronal damage [19,58,78].

Our study has a series of limitations, which consist of the retrospective analysis of the data and the failure to observe changes in BMI over time.

## 5. Conclusions

Early diagnosis and treatment of MS have a very strong impact on the prognosis of multiple sclerosis. The identification of risk factors for the development of the disease is of great importance. Our study strengthens the thesis that being overweight may have an unfavorable prognostic role in MS patients. The onset of MS in childhood offers disadvantages in terms of impact on cognition, future disability, and reduced availability of drugs to reduce disease progression. To these, the addition of overweight in childhood may have a further unfavorable prognostic factor. Controlling weight during adolescence, rather than childhood or adulthood, is critical in determining the risk of MS.

## Figures and Tables

**Figure 1 nutrients-15-04880-f001:**
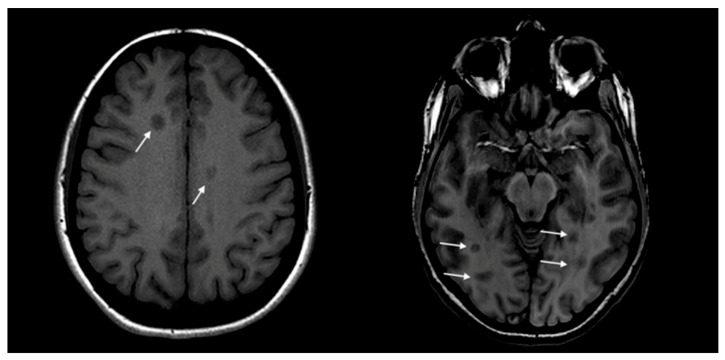
Onset MRI scans of two enrolled patients. The white arrows indicate T1 hypointense lesions, known as black holes.

**Table 1 nutrients-15-04880-t001:** Demographic features of the patients included in the study. SD: standard deviation.

	POMS Subjects			
Sex	No.	%	Age, y	
			Mean	SD
Male	21	38.2%	13.1	3.2
Female	34	61.8%	13.7	3.06
Total	55	100%	13.5	3.1

**Table 2 nutrients-15-04880-t002:** Demographic features of the healthy weight and overweight/obese patients.

	Healthy Weight				Obese/Overweight			
	No.	%	Age, y		No.	%	Age, y	
			Mean	DS			Mean	DS
Total	33	60	14.6	4.1	22	40	12.7	3.8
Male	14	42.4	12.4	3.5	7	31.8	14.5	1.8
Female	19	57.6	13	3.8	15	68.2	14.6	1.2

**Table 3 nutrients-15-04880-t003:** Clinical features at onset of the healthy weight and overweight/obese patients.

	Healthy Weight		Obese/Overweight		*p*-Value
	Mean	SD	Mean	SD	
Age at onset, y	14.6	4.1	12.7	3.8	<0.05
Expanded disability status scale (EDSS) at onset	1.9	0.5	2.2	0.6	0.07
	No.	%	No.	%	
Monofocal onset	22	66.6	4	18.1	<0.05
Polifocal onset	7	21.2	16	72.7	<0.05
Encephalopathy at onset	4	12.1	2	9.1	0.08
Pyramidal functions	8	25.4	11	50	<0.05
Superficial sensation (light, touch and pain)	9	27.3	11	50	0.07
Proprioceptive sensation	9	27.3	6	27.3	0.08
Cerebellar functions	9	27.3	3	13.6	0.19
Brainstem functions	16	48.5	9	40.9	0.39
Visual deficit	10	30.3	9	40.8	0.3
Bowel and bladder functions	2	6.1	2	9.1	0.52
Oligoclonal band (intrathecal IgG synthesis)	26	83.9	19	95	0.23
Pleocytosis (>5 cell/mmc)	13	50	14	70	0.14
Ig G anti-Epstein–Bar virus	30	91	22	100	0.2

**Table 4 nutrients-15-04880-t004:** Onset Magnetic Resonance Imaging features for the healthy weight and overweight/obese patients.

	Healthy Weight		Obese/Overweight		*p*-Value
	No.	%	No.	%	
Black holes	5	15.5	12	54.5	<0.05
Periventricular lesion	29	87.9	18	81.8	0.4
Juxtacortical/cortical	25	75.8	19	86.4	0.27
Infratentorial	23	69.7	13	59.1	0.3
Optic nerve	13	39.4	9	40.9	0.5
Spinal cord	20	60.6	13	59.1	0.09
Gadolinium enhancing lesions	25	75.8	19	86.4	0.27
Tumefactive lesions	8	24.2	6	27.3	0.52

## Data Availability

The data presented in this study are available on request from the corresponding author. The data are not publicly available due to ethical reasons.

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
