# Peer review of "Pediatric Onset Multiple Sclerosis and Obesity: Defining the Silhouette of Disease Features in Overweight Patients"

_nutrients, 2023, doi:10.3390/nu15234880_

Round 1
Reviewer 1 Report
Comments and Suggestions for Authors
Dears,
Manuscript entitled "Pediatric onset multiple sclerosis and obesity: defining the silhouette of disease features in overweight patients" is an original article addressing the correlation between obesity / overweight and the clinical picture at the onset of multiple sclerosis (MS) in children. The article regards an important risk factor of MS - obesity during childhood.
There are some important issues that require correction:
- lack of control group
- studied groups are rather small - 33 and 22 patients
- methods are not clearly explained, there is no explanation how BMI was determined, what was MRI field strength
- in Table 3 clinical features should be better described, e.g. pyramidal syndrome/signs, superficial sensation disturbances, cerebellar syndrome or signs, oligoclonal bands presence in the CSF, pleocytosis i.e. number of cells in the CSF > 5/ul
- previous Epstein Barr Virus infection should be defined - whether in history or determined by the presence of antibodies
- children are addressed as women and men, it should be rather boys and girls.
Besides, there are a lot of grammar, style and spelling mistakes that require correction by a native speaker.
Comments on the Quality of English Language
There are a lot of grammar, style and spelling mistakes that should be corrected by a native speaker.
Author Response
- Lack of control group. The design of our study includes a study group which is overweight patient with multiple sclerosis (MS) and a control group which is non-overweight MS patients. It is difficult to think of another control group such as obese patients without multiple sclerosis because they are not suitable for the purpose of the study: "to understand if obese multiple sclerosis patients have a more severe form of the disease than normal weight multiple sclerosis patients.
- Size of the group. Multiple sclerosis in childhood is a rare condition, with pooled global prevalence was calculated to be 8.11 (95% CI: 2.28–13.93) per 100,000 people (ref. https://doi.org/10.1016/j.msard.2020.102260).
this explains why the number of patients in the study is low. Below are references to works published in journals of high scientific impact which confirm that we are often forced to report data on a low number of subjects with pediatric MS.
Below are some references as examples.
10.1016/j.msard.2023.104970
10.1016/j.ejpn.2023.05.001
10.1080/21622965.2022.2082874
10.1177/13524585221144742
10.1016/j.msard.2023.104502
- Methods. Regarding MRI, this is not a study aimed at investigating new MRI techniques but involves the use of MRI for the evaluation and characterization of lesions according to the standardized protocol of the "Magnetic Resonance Imaging in Multiple Sclerosis (MAGNIMS) study group". The sequences included in this protocol are those described in the methods section. There are no innovative techniques for the purpose of this study, but if the reviewer has something specific in mind he can tell us.
- The clinical characteristics of the symptoms presented by patients are classified according to the Expanded Disability Status Scale (EDSS) system and shared by multiple sclerosis experts internationally. Within each sphere there is a very numerous spectrum of symptoms whose list has no repercussions on the classification system or on the diagnosis or prognosis. Instead, we added the suggested specifications for oligoclonal bands and pleocytosis, although they had already been specified in detail in the materials and methods section.
- In the methods section it is clearly stated that for previous EBV infection we referred to the positivity of anti-Ig G antibodies against the virus.
- We have replaced the terms man and woman with boy and girl.
- English was reviewed by a native speaker
Reviewer 2 Report
Comments and Suggestions for Authors
The manuscript entitled "Pediatric onset multiple sclerosis and obesity: defining the silhouette of disease features in overweight patients" brings an interesting hypothesis regarding the pathogenesis of multiple sclerosis, considering obesity as a risk factor for BB disruption due to proinflammatory cytokines produced by adipose tissue. Would be very interesting to documentate if there is any connection between proinflammatory molecules produced by adipose tissue and the various clinical forms of MS. Otherwise, the hypotheses of adipose tissue induced BB disruption is unverified and is can be consider it only a supposition.
The present manuscript needs to add the specification regarding the Ethic Committee Approval (of the hospital where the study was made) and the informed consent for the each patient's parents.
Author Response
We thank the reviewer for the suggestion which we will capitalize on for the next research. However, our work did not examine the study of inflammatory markers of adipocyte origin (for example in the blood). It is a retrospective study that analyzed only clinical data. We therefore agree that it remains a supposition and we underlined this in the discussion.However, it could be an interesting fact that there are no differences between the frequency of presence of oligoclonal bands in the CSF of overweight patients compared to normal weight patients or in the link index which can be two measures of barrier damage.
Specifications on the ethics committee have been added in the methods-subjects section
Reviewer 3 Report
Comments and Suggestions for Authors
The manuscript is focused on pediatric onset of multiple sclerosis phenotypes in relation to obesity. Onset age, monofocal onset, polyfocal onset, pyramidal function and black holes in MRI have been significantly associated with overweight/ obesity in children with POMS which support the higher risk of MS phenotypes in obese children with POMS.
Similar studies on retrospective data of ill children are very valuable for clinical practice. The article is well written („polifocal onset“ 131/4?).
Author Response
Thanks for your comments.
Round 2
Reviewer 1 Report
Comments and Suggestions for Authors
After corrections made by the authors, the manuscript can be published. It still requires editing of English language.
Comments on the Quality of English LanguageThe manuscript still requires editing of English language.